# Liver-Derived Exosomes and Their Implications in Liver Pathobiology

**DOI:** 10.3390/ijms19123715

**Published:** 2018-11-22

**Authors:** Sumi Sung, Jieun Kim, Youngmi Jung

**Affiliations:** 1Department of Integrated Biological Science, Pusan National University, 63-2 Pusandaehak-ro, Kumjeong-gu, Pusan 46241, Korea; sungsm06@pusan.ac.kr (S.S.); jieun@pusan.ac.kr (J.K.); 2Department of Biological Sciences, Pusan National University, 63-2 Pusandaehak-ro, Kumjeong-gu, Pusan 46241, Korea

**Keywords:** exosome, extracellular vesicles, liver, hepatocytes, nonparenchymal cells, liver disease

## Abstract

The liver has a wide range of physiological functions in the body, and its health is maintained by complex cross-talk among hepatic cells, including parenchymal hepatocytes and nonparenchymal cells. Exosomes, which are one method of cellular communication, are endosomal-derived small vesicles that are released by donor cells and delivered to the target cells at both short and long distances. Because exosomes carry a variety of cargoes, including proteins, mRNAs, microRNAs and other noncoding RNAs originating from donor cells, exosomes convey cellular information that enables them to potentially serve as biomarkers and therapeutics in liver diseases. Hepatocytes release exosomes to neighboring hepatocytes or nonparenchymal cells to regulate liver regeneration and repair. Nonparenchymal cells, including hepatic stellate cells, liver sinusoidal endothelial cells, and cholangiocytes, also secrete exosomes to regulate liver remodeling upon liver injury. Exosomes that are released from liver cancer cells create a favorable microenvironment for cancer growth and progression. In this review, we summarize and discuss the current findings and understanding of exosome-mediated intercellular communication in the liver, with a particular focus on the function of exosomes in both health and disease. Based on the current findings, we suggest the potential applications of exosomes as biomarkers and therapeutics for liver diseases.

## 1. Introduction

Exosomes are extracellular vesicles (EVs) that are cell-derived membranous structures [1]. The secretion of EVs was initially described as a method for eliminating unneeded compounds from cells [2]. However, exosome have been shown to act as signaling vehicles for cell-to-cell communication because cells exchange bioactive components, including nucleic acids and proteins, via exosomes through delivery into the cytoplasm of recipient cells [3]. Diverse body fluids, such as saliva, serum, plasma, and urine, contain numerous exosomes carrying a variety of proteins, mRNAs microRNAs (miRs) and other noncoding RNAs [4,5,6,7,8,9]. Cells produce exosomes under both normal and pathological conditions, and the components or numbers of exosomes secreted by cells vary, depending on the conditions of donor cells [10,11,12]. Because of the capability of exosomes to act as carriers, exosomes have been recognized as possessing clinical applications as both diagnostic biomarkers and therapeutic tools [13]. Therefore, they are now considered to have strong potential to contain valuable biomarkers or exhibit therapeutic effects on experimental and clinical conditions.

The liver, which is the largest internal organ of the human body, is a functionally complex organ that plays critical physiological roles in metabolism, detoxification, digestion, synthesis, and storage [14,15,16,17]. Parenchymal cells, also known as hepatocytes, occupy approximately 80% of the total liver volume and perform the majority of numerous liver functions [18]. Nonparenchymal cells include hepatic stellate cells (HSCs), liver sinusoidal endothelial cells (LSECs), hepatic macrophages (also called Kupffer cells), and cholangiocytes, and these cells only account for 6.5% of the liver volume, but 40% of the total number of liver cells [18]. These nonparenchymal cells release various substances to regulate or assist the functions of hepatocytes or neighboring nonparenchymal cells [19]. Specifically, liver cells release and/or receive exosomes for intercellular communication in both healthy and damaged livers [20]. Since alterations in the quantity and composition of exosomes reflect the pathophysiological status of donor cells, exosome-mediated intercellular communications are important in both normal liver physiology and disease processes. Based on emerging evidence, exosomes are crucial to the pathogenesis of various liver diseases, including liver cancer, viral hepatitis, nonalcoholic fatty liver disease (NAFLD), and alcoholic liver disease (ALD) [20,21]. In addition, exosomes have recently been suggested to serve as an ideal candidate for a novel diagnostic biomarker or an effective therapeutic tool [20,22]. However, few studies have examined liver exosomes, since most of the research has focused on therapeutic effects of mesenchymal stem cells (MSCs)-derived exosomes [23]. This review summarizes the current understanding and findings on liver cell-derived exosomes and highlights their theragnostic (diagnostic and therapeutic) potential. Liver cells, including hepatocytes, HSCs, LSECs, and cholangiocytes, have been reported to be exosome-releasing donor cells and exosome-receiving recipient cells both in health and in disease states. Hence, information about the mechanisms by which liver cells communicate via exosomes and what cargoes are carried within exosomes from specific donor cells will improve our understanding of liver pathophysiology.

## 2. Exosomes

In addition to the release of secretory vesicles by specialized cells, which carry neurotransmitters or hormones, all cells are capable of secreting various types of membrane vesicles, known as EVs [24]. Two types of EVs have been characterized according to their origins: exosomes and microvesicles [25]. EVs are formed either by the budding of the plasma membrane, in which case they are referred to as microvesicles, or as intraluminal vesicles within the lumen of multivesicular endosomes (MVEs) [26,27]. MVEs fuse with the plasma membrane to release intraluminal vesicles that are then called exosomes [24]. Exosomes are small vesicles ranging in size from 30 to 100 nm that are secreted by all cell types and are present in most body fluids, including blood, saliva, and urine [4,28,29,30]. Initially, the proposed role for exosomes is to remove cellular waste, but many researchers have reported that exosomes are more than just waste carriers and they are involved in biological processes [2,24]. Exosomes function both in short-range intercellular communication when taken up by adjacent cells and in long-range communication when released into the bloodstream [31]. Hence, exosomes are now considered to be another member of the cellular secretome that facilitates intercellular communication and provides information about the parental cells, and the main focus of the field is now their capacity to exchange components between cells.

Exosomes encapsulate their cargoes in a lipid bilayer-bound vesicle similar to the plasma membrane of cells [32]. Exosomes are highly heterogeneous, depending on the cell type, but contain common “exosome marker” proteins that are associated with exosome biogenesis and secretion, and cell-type specific proteins and nucleic acids that reflect the phenotypic state of the cell from which they are generated [27]. The lipid composition of exosomes includes cholesterol, sphingomyelin, hexosylceramides, phosphatidylserine, and saturated fatty acids [24]. Since they originate from the endosomal system, proteins enriched in exosomes include proteins that are associated with membrane transport and fusion, including Rab (Ras-related proteins in brain) GTPases and annexins, as well as proteins involved in exosome biogenesis, including endosomal sorting complexes that are required for transport (ESCRT) complexes, ALG-2-interacting protein X (ALIX), and tumor susceptibility gene 101 (TSG101) [24]. Exosomes are also enriched with heat shock proteins (HSP60, HSP70, and HSP90), integrins, and tetraspanins (CD9, CD10, CD26, CD63, CD81, and CD82) [20,24,27]. Although many exosomal protein constituents likely reflect the common exosome biogenesis pathway, other proteins may be enriched in exosomes as a result of enhanced levels in the parental cell [10]. In addition, mRNAs and miRs are the primary nucleic acids that are identified within exosomes [33]. Other RNA species, including noncoding RNAs, transfer RNAs, mitochondrial RNA, and single-stranded DNA, have also been detected in the lumen of exosomes [34,35,36]. These multiple exosomal cargoes within exosomes are transferred to the recipient cell and trigger intracellular signaling pathways, eventually regulating cellular events in the recipient cell [37]. Although the functional impacts of exosomal cargoes for intercellular communication are still emerging, each cell type tunes exosomes depending on its physiological state and releases exosomes with particular lipid, protein, and nucleic acid compositions [38]. Depending on the donor cell types or conditions, contents of the exosomes can vary widely, and a distinct set of cell-type-specific cargoes can influence recipient cells and alter the surrounding microenvironments. Thus, exosomes are another member of the cellular secretome that function as a paracrine/systemic messenger affecting other cells and providing information about the parental cells’ conditions.

## 3. Exosomes in the Liver

Although hepatocytes perform the majority of liver functions, the remaining population of nonparenchymal cells, including LSECs, HSCs, and cholangiocytes, is also necessary for liver function and pathology [39]. Nonparenchymal cells release various substances to regulate or assist the functions of hepatocytes, as well as to perform the respective functions of nonparenchymal cells [19]. Exosomes are considered to be important mediators of cell-cell communication by delivering the information that they carry to neighboring cells [3]. In liver pathophysiology, multiple interactions or methods of cross-talk between both identical cell types or between different cell types is an inevitable process through exosomes or other mechanisms [40]. Therefore, knowledge of how liver cells communicate via exosomes and the functions of cell-type specific exosomes from each cell type will facilitate studies of liver disease. Here, we review and discuss the current knowledge of how liver cells interact with other cells via exosomes and how exosomal cargoes affect many signaling events among multiple hepatic cell types. The contents and actions of the exosomes that are produced in the liver are summarized in the Figure 1, according to the type of cells secreting the exosomes.

### 3.1. Hepatocyte-Derived Exosomes

Hepatocytes, which are a major type of hepatic cells, occupy approximately 80% of the total liver volume and perform the major liver functions of glycolysis, bile production, detoxification, and regeneration [14,41,42]. Although hepatocytes remain in a quiescent state, hepatocytes enter the cell cycle and proliferate to repopulate damaged regions when the healthy liver is injured [43]. However, when liver damage is severe, hepatocytes are unable to divide and replace damaged tissues [42]. Then, dying or ballooned hepatocytes release a large number of chemokines/cytokines and provoke the participation of nonparenchymal cells in liver reconstitution [44,45]. Hence, hepatocytes must maintain their regenerative capacity and deliver the signals that stimulate nonparenchymal cells.

Hepatocytes communicate with other hepatocytes (hepatocyte-to-hepatocyte communication) via exosomes to promote liver repair and regeneration after injury. Exosomes that are released by hepatocytes fuse with and promote the proliferation of target hepatocytes. Nojima et al. [46] reported that exosomes derived from hepatocytes delivered sphingosine kinase 2 (SK2) to form sphingosine-1-phosphate (S1P) in target hepatocytes. S1P formed by the SK2-catalyzed phosphorylation of sphingosine promoted cell survival, growth and migration, and promote hepatocyte proliferation in culture. When exosomes that were isolated from primary murine hepatocytes were administered to mice subjected to ischemia/reperfusion injury or partial hepatectomy, they induced intracellular synthesis of S1P within hepatocytes to promote cell proliferation. These findings were not observed when hepatocytes were treated with exosomes that were derived from Kupffer cells or LSECs, suggesting that SK2 might be a specific cargo in hepatocyte-derived exosome. Exosomes are also involved in viral spread during viral infections [47]. Cell co-culture studies revealed that hepatocytes infected with hepatitis C virus (HCV) have been shown to release exosomes containing the HCV RNA to infect the intact hepatocytes, indicating that exosome-mediated hepatocyte-to-hepatocyte communication contributes to HCV infection.

Hepatocytes also communicate with neighboring nonparenchymal cells via exosomes. HSCs, a nonparenchymal cell type, receive a distress signal from exosomes derived from damaged hepatocytes. Upon liver injury, damaged hepatocytes release exosomes, which are internalized into adjacent HSCs and induce transdifferentiation from quiescent (q) to activated (a)/myofibroblastic HSCs. According to Seo et al. [48], damaged hepatocytes produce exosomes harboring unknown ligands binding to toll-like receptor 3 (TLR3), which activates HSCs through TLR3 activation in the culture system. In vivo studies confirmed that activation of TLR3 in HSCs by exosomes derived from damaged hepatocytes exacerbates liver fibrosis by enhancing the production of chemokine (C-C motif) ligand 20 (CCL20) and interleukin-17A (IL-17A). Similarly, lipid-induced toxicity stimulates hepatocytes to release exosomes containing a fibrosis-inducing signal to HSCs. Lee et al. [12] reported that palmitic acid (PA)-treated hepatocytes displayed significantly increased exosome production, and treatment of HSCs with exosomes derived from PA-treated hepatocytes induced HSC activation in culture. Of their cargoes, exosomal miR-192 significantly increases the expression of profibrotic markers in HSCs. In addition, in vitro studies demonstrated that exosomes from HCV-infected hepatocytes carry miR-19a, which targets the suppressor of cytokine signaling 3 (SOCS3) to activate signal transducer and the activator of transcription 3 (STAT3)-mediated transforming growth factor beta (TGF-β) signaling pathway and activates HSCs [49]. Thus, healthy or injured hepatocytes release exosomes, and the released exosomes have important roles in both hepatocyte-to-hepatocyte communication and hepatocyte-to-HSC communication.

### 3.2. HSC-Derived Exosomes

HSCs are localized in the subendothelial space of Disse, interposed between LSECs and hepatocytes; they represent ~10% of all resident liver cells [18]. In the normal liver, HSCs are quiescent and they function as the major storage facility for retinoids (vitamin A) [17]. However, when the liver is injured, HSCs become activated/transdifferentiated from vitamin-A-storing cells to myofibroblasts and contribute to ECM accumulation by producing collagen fibrils [50]. Thus, HSCs are the major fibrogenic cells that promote the development of liver fibrosis. Since the nature and abundance of exosomal cargoes depend on the cell type and are influenced by the physiological or pathological state of the donor cell, specific stimuli that modulate their production and release lead to the biogenesis of exosomes. Among liver cell populations, HSC-derived exosomes well reflect the condition of donor cells and their compositions vary depending on the extent of liver injury.

Exosomes derived from qHSCs are known to possess anti-fibrotic properties, thus suppressing HSC activation. Chen et al. [51] detected that the level of miR-214 increased in qHSC-secreted exosomes as compared with aHSC-released exosomes, indicating that miR-214 possesses anti-fibrotic properties. Cell co-culture studies revealed that exosomal miR-214 produced by primary qHSCs is transferred intercellularly into primary aHSCs and it suppresses the expression of its direct target connective tissue growth factor (CTGF) and downregulates alpha-smooth muscle actin (a-SMA) and collagen expression downstream of CTGF. In addition, hepatocytes also take up qHSC-derived exosomes containing miR-214 to downregulate CTGF expression in recipient hepatocytes, indicating the exchange of exosomal miR-214 between HSCs and hepatocytes as well. The same group also reported that miR-199a-5p is an miRNA that is produced at a high level in exosomes derived from qHSCs and delivered to aHSCs in which fibrogenic gene expression, including CTGF, is then attenuated in which aHSCs were co-cultured with qHSCs [52]. On the other hand, exosomes that were derived from aHSCs play a role in promoting liver fibrosis. As primary HSCs are activated in culture, CTGF mRNA and protein levels increase and are also present in aHSC-derived exosomes [53]. CTGF is packaged into secreted exosomes, and exosomal CTGF is intercellularly delivered to other aHSCs or qHSCs, subsequently amplifying fibrogenic signaling in recipient HSCs. These findings suggest that the contents and composition of HSC-derived exosomes mainly depend on the fibrogenic state of HSCs, and HSCs communicate with other HSCs via exosomes to regulate the hepatic response to liver injury.

### 3.3. LSEC- or Cholangiocyte-Derived Exosomes

LSECs comprise approximately 50% of nonparenchymal cells in the liver and they are the first cells to contact the blood flow in hepatic sinusoids and transfer waste molecules to the circulation [54]. Given that LSECs are anatomically located near HSCs and release exosomes, exosomes that are derived from LSECs influence HSCs in a paracrine manner. Wang et al. [55] reported that LSECs release exosomes containing the sphingosine kinase 1 (SK1) protein; exosomal SK1 is delivered into HSCs and it promotes AKT phosphorylation, leading to HSC activation in vitro. Furthermore, in the mouse liver injured by bile duct-ligation (BDL) or carbon tetrachloride (CCl_4_) injection, the levels of SK1 were upregulated in both liver tissues and exosomes that were isolated from serum. However, treatment with an SK1 inhibitor protected mice from BDL-induced or CCl_4_-induced liver fibrosis. Thus, LSEC-derived exosomes, specifically the SK1 cargo, act as a primary mediator of HSC activation upon liver injury. In addition, Giugliano et al. [56] reported that HCV infection induces the production of type I/III interferons by LSECs, which stimulate LSECs in an autocrine manner to release exosomes with antiviral activity to prevent a new infection within LSECs and hepatocytes in vitro system. According to the data, exosomes are involved in the antiviral response during a viral infection.

Cholangiocytes represent approximately 3–5% of the endogenous liver cell population [57]. Although cholangiocytes occupy a small part of the liver, they have an important pathophysiological role in cholestatic liver injury [57]. Since cholangiocytes are epithelial cells that line a three-dimensional network of bile ducts, they are the primary target of cholestatic liver injury [58]. Cholangiocytes are normally quiescent in the liver, but the accumulation of bile acids in the damaged liver activates the inflammatory response, resulting in the destruction of intrahepatic bile ducts and eventually leading to cholangitis, fibrosis, and potentially cirrhosis [59]. Cholangiocytes are exosome-releasing cells. Li et al. [60] detected the enrichment of the long noncoding RNA H19 in exosomes that were collected from serums from patients with primary sclerosing cholangitis and CCl_4_-injected mice, as well as cholangiocyte-derived exosomes under cholestatic conditions. They presented that exosomal H19 was delivered into hepatocytes and suppressed small heterodimer partner (SHP) expression in both in vivo and vitro systems, leading to hepatic inflammation and severe cholestatic liver injury.

### 3.4. Liver Cancer Cell-Derived Exosomes

In the tumor microenvironment, intercellular communication (e.g., cross-talk between cancer cell and stromal cells) plays key roles in promoting tumor progression and metastasis [61]. Cancer cells secrete more exosomes than healthy cells [62]. Cancer cells have recently been shown to modulate the surrounding hepatic environment to aid in their growth, proliferation, and invasion. HCC cells secrete exosomes that contain a variety of cargoes, which might contribute to intercellular communication. Kogure et al. [63] performed exosomal miRNA profiling in donor HCC cells in vitro and identified that miRNAs contained within exosomes modulate gene expression and cell signaling related to cancer cell growth (e.g., transforming growth factor beta activated kinase-1 (TAK1) signaling) in recipient HCC cells. HCC cell-derived exosomes also promote the migration and invasion of HCC cells by stimulating recipient hepatocytes to secrete matrix metalloproteinase-2 and -9 that facilitate the invasion of HCC cells. In addition, exosomes that are derived from the metastatic HCC cell line HEP3B induce the migration and invasion of immortalized MIHA hepatocytes by transferring oncogenic mRNAs or proteins, such as caveolins and members of the S100 family, which trigger PI3K/AKT and MAPK signaling in recipient hepatocytes [64]. Similar results have also been observed in studies investigating the activation of cancer-associated fibroblasts [65]. Notably, miR-1247-3p derived from highly metastatic HCC cells is directly transferred to fibroblasts via exosomes and converts fibroblasts to cancer-associated fibroblasts by decreasing the expression of its target beta-1,4-galactosyltransferase 3 (B4GALT3) and activating NF-κB signaling in fibroblasts. These activated fibroblasts promote the stemness, epithelial-to-mesenchymal transition, chemoresistance, and tumorigenicity of liver cancer cells by secreting IL-6 and IL-8. Hence, tumor-derived exosomal miR-1247-3p has an important role in intercellular communication to foster an inflammatory and tumor-favorable microenvironment. Thus, cancer cells cooperate with other cancer cells to promote their growth, migration, and invasion, or even noncancer cells to remodel the surrounding parenchymal tissue and support tumor progression via exosomes by modifying the microenvironment.

## 4. Applications of Exosomes in Liver Diseases

Because exosomes contain donor cell-specific proteins and nucleic acids that reflect the physiological health or pathophysiological state, they hold potential as platforms for biomarker discovery [66]. Furthermore, the ability of exosomes to communicate among cells locally and systemically has prompted evaluations of repurposing these vesicles as carriers of therapeutic agents [67]. Hence, exosomes may serve as potentially useful biomarkers for diagnosis and prognosis, but may also serve as therapeutic targets in various liver diseases.

Liver biopsy is still the gold standard for diagnosing or determining the prognosis of liver diseases, although the procedure has several limitations for the diagnosis of liver diseases [68]. Because only a small portion of the whole liver tissue is sampled during liver biopsy, despite the use of a thick needle or the collection of two or more samples, the various stages of hepatic fibrosis are difficult to diagnose accurately [68]. Integrated criteria to diagnose nonalcoholic steatohepatitis (NASH) are unavailable, and criteria that are based on the various histopathological features are not sufficient to distinguish AFLD/NAFLD, since they share similar histopathological features [69]. In addition, liver biopsy is an invasive procedure with associated morbidity: 20% of patients experience pain and 0.5% of patients experience major complications (such as bleeding or hemobilia) [70]. Therefore, noninvasive methods, such as biochemical and/or hematological tests based on biomarkers, have been reported to have an advantage over liver biopsy to assess the degree of liver damage. Exosomes present in blood and urine contain specific proteins, mRNAs and miRNAs derived from multiple organs, including the liver, which have been considered predictive biomarkers [71]. In addition, exosomes have much longer half-lives than proteins, RNAs, and other molecules [72]. Therefore, the characterization of cell-specific exosomes represents a potentially useful option for diagnosing or determining the prognosis of liver diseases. Exosomal cargoes showing differential expression between healthy and disease states (e.g., the absence/presence of fibrosis) might be novel candidate noninvasive biomarkers of liver disease. For example, CTGF, a fibrosis-related cargo that we discussed in this review, has potential utility as a noninvasive biomarker of liver fibrosis because the levels of CTGF are substantially increased in fibrotic liver tissues or serum that were collected from human patients [73]. In addition, other fibrosis-related cargoes, such as miR-214, miR-199a-5p, and CTGF, may have potential utility as noninvasive biomarkers of liver fibrosis because their levels differ according to the presence of fibrosis [51,52]. However, levels in circulating exosomes do not exclusively reflect the production of exosomes from the liver because serum exosomes contain numerous exosomes that are released from multiple organs. Therefore, the identification of an organ-specific marker in exosomes is important for the application of exosomes as noninvasive biomarkers. Lee et al. [12] also proposed that the exosomal miR-192 from damaged hepatocytes represents a potential biomarker for NAFLD progression from simple steatosis to NASH. The levels of miR-192 were increased in circulating exosomes in sera from patients with advanced stage NAFLD as compared to patients with early stage NAFLD, suggesting that miR-192 represents a potential biomarker that is associated with NASH progression. However, further investigations are required to identify the mechanism of exosomal miR-192 production and the role of miR-192 in NASH progression.

Many researchers have focused their attention on the therapeutic effects of MSCs due to the immunosuppressive and anti-inflammatory properties of MSCs. The therapeutic efficacy of MSC-derived exosomes has been reported in a variety of animal models of liver disease. According to Li et al. [74], exosomes that are derived from human umbilical cord MSCs ameliorate liver fibrosis by inhibiting the hepatocyte EMT and collagen production. Hyun et al. [75] also showed that exosomes derived from chorionic plate-derived MSCs contain miR-125b, which alleviates hepatic fibrosis by abrogating hedgehog (Hh) signaling and inhibiting HSC activation. Exosomes that are derived from adipose-derived MSCs contain miR-181-5p, and this miRNA ameliorates liver fibrosis by suppressing HSC activation through the direct targeting of Bcl-2 and STAT3 [76]. Hence, the ability of exosomes to transport cargoes with therapeutic efficacy has attracted increasing attention. By replacing the transplantation of MSCs with the administration of their secreted exosomes, many of the safety concerns and limitations that are associated with the transplantation of viable replicating cells might be mitigated. For example, the use of viable replicating cells as therapeutic agents carries the risk that biological potency of the agent may persist or is unable to be attenuated after treatment is terminated [77]. In addition, their potential to differentiate into hepatocytes or myofibroblasts might lead to adverse outcomes if treatment is terminated. Therefore, exosomes, a cell-free agent, circumvent some of the challenges that are associated with the use of MSCs by exhibiting lower immune responses, an inability to directly form tumors and safe storage without losing function.

While extensive research is currently underway to assess the use of MSC-derived exosomes, few studies on the use of liver cell-derived exosomes are available. According to Nojima et al. [46], exosomal SK2 isolated from hepatocyte-derived exosomes promoted hepatocyte proliferation and liver regeneration, suggesting that exosomes that are derived from normal hepatocytes exert a positive effect on liver regeneration. Exosomal miRs derived from quiescent HSCs exert a therapeutic effect to relieve liver fibrosis. Exosomal miR-214 or miR-199a-5p isolated from qHSC-derived exosomes induce a therapeutic effect on liver fibrosis by inhibiting HSC activation [51,52]. Hence, healthy liver cell-derived, but not MSC-derived, exosomes have therapeutic potential as a treatment for liver disease. Furthermore, another therapeutic strategy is to block or interrupt exosome-mediated intercellular communication by inhibiting exosome release from donor cells, such as damaged hepatocytes, aHSCs, activated LSECs, or cholangiocytes. For example, the blockade of exosome release by damaged hepatocytes reduces liver fibrosis by interrupting the transduction of fibrosis-inducing signals to HSCs [49,78]. However, the mechanisms that control the sorting of molecules into exosomes and the release of exosomes by specific stimuli remain unclear. Further studies of these processes will provide new insights into and overcome the limitations for the use of exosomes in therapeutic applications.

## 5. Conclusions and Perspectives

Exosome-mediated cell-cell communication modulates various cellular functions by transferring a variety of bioactive components from donor cells to target cells [3]. In the liver, hepatocytes and nonparenchymal cells, including HSCs, cholangiocytes, and LSECs, may not only be the origin of the exosomes, but are also potentially targets themselves both in health and disease states. Thus, liver cells communicate with other cells via exosomes and eventually regulate cellular events in the recipient cell. Therefore, an understanding of how liver cells communicate via exosomes will be important. Furthermore, exosomes may serve as useful biomarkers for diagnosis and prognosis, as well as therapeutic targets in liver diseases. Despite the great potential of exosomes, several important issues still remain. Exosome isolation techniques should be improved before their experimental and clinical applications, because these processes are time-consuming and the yield is too low [79]. Additionally, very little is known about the precise molecular mechanisms of exosome biogenesis, release, targeting, and interactions with target cells, particularly in the liver. Markers for organ-specific or cell-specific exosomes must be identified to clarify their functions, depending on their origin.

## Figures and Tables

**Figure 1 ijms-19-03715-f001:**
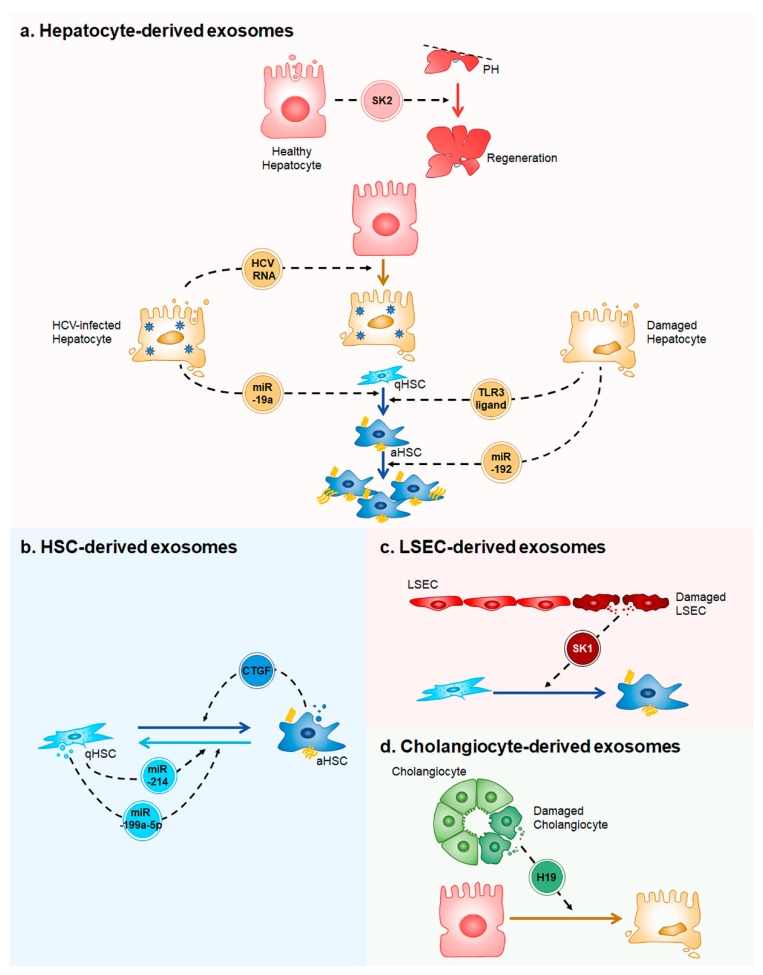
A simplified model of exosome-mediated intercellular communication in the liver. (**a**) Healthy hepatocytes release exosomes containing sphingosine kinase 2 (SK2), which promotes liver regeneration in two-thirds partial hepatectomied (PH) liver. Hepatitis C Virus (HCV)-infected hepatocytes secrete exosomes having HCV RNA to infect the uninfected hepatocytes. MicroRNA (miR)-19a in exosomes derived from the HCV-infected hepatocytes induces trandifferentiation of quiescent hepatic stellate cells (qHSCs) into activated HSCs (aHSCs). When liver is injured, damaged hepatocytes release exosomes containing unknown toll-like receptor 3 (TLR3)-ligands or miR-192. The ligands lead to HSC activation and miR-192 promotes proliferation of aHSCs; (**b**) Exosomes containing miR-214 or miR-199a-5p produced by qHSCs inhibit HSC activation, whereas exosomes having connective tissue growth factor (CTGF) released from aHSCs stimulate HSC activation; (**c**) Injured LSECs release exosomes containing sphingosine kinase 1 (SK1), which induces HSC activation; and, (**d**) Damaged cholangiocytes secrete exosomes containing long non-coding RNA H19 (H19). H19 injuries hepatocytes.

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
