# Peer review of "Liver-Derived Exosomes and Their Implications in Liver Pathobiology"

_ijms, 2018, doi:10.3390/ijms19123715_

Reviewer 1 Report

I read with interest the manuscript by Sung et al, dealing with the emerging role of liver-derived exosomes in physiology and pathology of the liver.

In general, the article does not add novelty: during the current and previous year different review articles have been written on this topic. As an example, the authors did not cite the recent review by Guo et al (Exosome: An Emerging Participant in the Development of Liver Disease, Hepat Mon 2017).

The article is well written and organized, however I have some minor concerns.

Title: Liver-derived exosomes and their implications in liver. It seems like a word is missing after "liver": maybe the authors intended "liver function"?. Please correct

The sections 1,2 3 contain repetitions. In particular authors reported in several occasions general concepts (e.g.: line 107 and lines115-119 seem to be repetition of Introduction section contents).  Please avoid repetitions

The manuscript does not include figures which could be helpful to better understand the complex interaction between exosomes and liver cells. Please add.

Author Response

In general, the article does not add novelty: during the current and previous year different review articles have been written on this topic. As an example, the authors did not cite the recent review by Guo et al (Exosome: An Emerging Participant in the Development of Liver Disease, Hepat Mon 2017).

We cited this review [Gu et al. Hepat Mon. 2017] (line 121 on page 3) in the revised manuscript.

The article is well written and organized, however I have some minor concerns.

Title: Liver-derived exosomes and their implications in liver. It seems like a word is missing after "liver": maybe the authors intended "liver function"?. Please correct

Dysregulated function of liver includes distortion in liver structure. In cellular level, transdifferentiation of quiescent HSCs into activated HSCs also accompanies morphological change, in addition to functional change. Hence, we intentionally write liver, not liver function. We hope you would accept author’s intention.

The sections 1,2 3 contain repetitions. In particular authors reported in several occasions general concepts (e.g.: line 107 and lines115-119 seem to be repetition of Introduction section contents).  Please avoid repetitions

Thank you for your helpful comment. As your requests, we deleted the repeated parts (line 107, line 112-118 on page 3) which were already described in introduction section.

The manuscript does not include figures which could be helpful to better understand the complex interaction between exosomes and liver cells. Please add.

As you requested, we changed the table with the figure in the revised manuscript. The figure depicts the liver cells-derived exosomes and exosome-mediated intercellular interaction.

Reviewer 2 Report

The paper represent a comprehensive overview on the research on hepatic exosomes.

There two points to consider:

1. It would be helpful in order to classify the different findings to state which data came from human and which one from murine studies, and if some of the findings which were made in tissue cultures or animal studies have been supported by human data.

2. On page 4 the authors mentioned the possibility of HCV transmission from cell to cell via exosomes. The better wording here would be contribute to the HCV-Infection rather than are responsible for the transmission.

Author Response

1. It would be helpful in order to classify the different findings to state which data came from human and which one from murine studies, and if some of the findings which were made in tissue cultures or animal studies have been supported by human data.

Thank you for your helpful comments. We clarified them in the revised this manuscript. (line 162, line 171, line 178, line 179, line 207, line 227, line 236, line 240)

2. On page 4 the authors mentioned the possibility of HCV transmission from cell to cell via exosomes. The better wording here would be contribute to the HCV-Infection rather than are responsible for the transmission.

As your comment, we changed ‘are responsible for viral infections’ with ‘contributes to the HCV-infection’ (line 165 on page 4 in the revised manuscript).
